# Physical Characterization and In Vitro Toxicity Test of PDMS Synthesized from Low-Grade D4 Monomer as a Vitreous Substitute in the Human Eyes

**DOI:** 10.3390/jfb13010003

**Published:** 2022-01-02

**Authors:** Diba Grace Auliya, Soni Setiadji, Fitrilawati Fitrilawati, Risdiana Risdiana

**Affiliations:** 1Department of Physics, Faculty of Mathematics and Natural Sciences, Universitas Padjadjaran, Jl. Raya Bandung-Sumedang km 21 Jatinangor, Sumedang 45363, Indonesia; fitrilawati@phys.unpad.ac.id; 2Department of Chemistry, Faculty of Mathematics and Natural Sciences, Universitas Padjadjaran, Jl. Raya Bandung-Sumedang km 21 Jatinangor, Sumedang 45363, Indonesia; s.setiadji@uinsgd.ac.id; 3Department of Chemistry, Faculty of Sciences and Technology, UIN Sunan Gunung Djati Bandung, Jl. A.H. Nasution No. 105, Cibiru, Bandung 40614, Indonesia

**Keywords:** HET-CAM, high surface tension, low-grade D4, polydimethylsiloxane, toxicity

## Abstract

Polydimethylsiloxane (PDMS) is one of the most superior materials and has been used as a substitute for vitreous humor in the human eye. In previous research, we have succeeded in producing PDMS with low and medium viscosity using octamethylcyclotetrasiloxane (D4) monomer with a low grade of 96%. Both have good physical properties and are comparable to commercial product PDMS and PDMS synthesized using D4 monomer with a high grade of 98%. An improvement of the synthesis process is needed to ensure that PDMS synthesized from a low-grade D4 monomer under specific synthesis conditions can repeatedly produce high-quality PDMS. Apart from good physical properties, the PDMS as a substitute for vitreous humor must also be safe and not cause other disturbances to the eyes. Here, we reported the process of synthesizing and characterizing the physical properties of low- and medium-viscosity PDMS using a low-grade D4 monomer. We also reported for the first time the in vitro toxicity test using the Hen’s Egg Test Chorioallantoic Membrane (HET-CAM) test method. We have succeeded in obtaining PDMS with viscosities of 1.15 Pa.s, 1.17 Pa.s, and 1.81 Pa.s. All samples have good physical properties such as refractive index, surface tension, and functional groups that are similar to commercial PDMS. The HET-CAM test results showed that all samples did not show signs of irritation indicating that samples were non-toxic. From the results of this study, it can be concluded that PDMS synthesized from a low-grade D4 monomer under specific synthesis conditions by the ROP method is very safe and has the potential to be developed as a substitute for vitreous humor in human eyes.

## 1. Introduction

Polydimethylsiloxane (PDMS) is one of the most superior materials and is commonly used to replace the vitreous humor damaged in the human eye through vitreoretinal surgery. Low-viscosity PDMS are preferred in vitreoretinal surgery because they are easier and faster to inject and expel than high-viscosity PDMS. However, PDMS with low viscosity also has a weakness in the form of a higher possibility of emulsification than PDMS with high viscosity [1]. Impurities are one of the main reasons for emulsification [2,3]. These impurities are thought to cause ocular toxicity by spreading to the surrounding eye tissue. Medium-viscosity PDMS is present as a new type of PDMS that offers advantages where the material is easier to inject than PDMS with high viscosity and has a lower emulsification tendency than other types of PDMS [4].

PDMS is synthesized from the octamethylcyclotetrasiloxane (D4) monomer. According to the European Chemical Agency (ECHA), D4 monomer is a toxic substance [5]. Several studies reported that D4 monomer has been shown to cause severe inflammation in rabbit and human eyes by penetrating ocular tissue [5,6,7]. D4 monomer caused acute ocular toxicity in the form of severe corneal edema and opacification [6]. Therefore, products produced from D4 monomer may contain hazardous and toxic substances. PDMS as a vitreous substitute will be used in human eyes and it must be non-toxic. For this reason, it is necessary to test the toxicity of PDMS to ensure the safety of using PDMS as a substitute for vitreous.

Various tests to replace rabbits in detecting potential chemical irritants have been developed [8]. One of them is the chicken egg chorioallantoic membrane (CAM) developed to detect a chemical test called the Hen’s Egg Test Chorioallantoic Membrane (HET-CAM) assay [9]. The existence of the HET-CAM test can be an initial test for eye irritation tests so that the use of animals in toxicological tests can be reduced [8]. CAM is a complete set of arteries, capillaries, and veins. CAM is technically easy to learn. Changes in vascular injury in the form of bleeding, lysis, and coagulation respond to certain irritant chemicals [10]. Previous studies have reported a good correlation between in vitro tests with HET-CAM and in vivo tests with the Draize method. The HET-CAM test has advantages over other tests due to its speed, simplicity, convenience, and relatively low cost [11].

Previous studies have reported that the synthesis of PDMS using low-grade D4 monomer has been successfully carried out by the ring-opening polymerization (ROP) method and produced low-viscosity PDMS with properties similar to commercial products and PDMS synthesized using high-grade D4 monomer [12]. Medium viscosity PDMS has also been successfully synthesized using the same monomer by optimizing the synthesis parameters [13]. However, it is necessary to improve the synthesis process by repeating the synthesis to ensure that the PDMS synthesized from a low-grade D4 monomer under these synthesis conditions has high-quality PDMS. In addition, further information, especially the level of toxicity of PDMS synthesized from low-grade D4 monomer, is also unknown. The test is essential considering that PDMS will be used in the human body to guarantee its safety. Our study focuses on how to produce high-quality PDMS from low-grade D4 monomer in the terms of its viscosity, refractive index, surface tension, material content, and safety suitable for use as a vitreous substitute. The synthesis process is an important part of achieving this goal. Here, we reported the synthesis and its physical characterization of low and medium-viscosity PDMS using a low-grade D4 monomer. We also reported for the first time the in vitro toxicity test of these PDMS samples using the HET-CAM test method to obtain information safety and toxicity level of all PDMS samples.

## 2. Materials and Methods

The low viscosity and medium viscosity of PDMS were synthesized by the ROP method. ROP is a chain-growth polymerization with the end of the polymer chain acts as a reactive center. The mechanism of ROP in the synthesis of PDMS is based on the cleavage of Si-O in the monomer used. D4 monomer with a high grade of 98% was also used as a comparison for low viscosity PDMS using D4 monomer with a low grade of 96%. The ROP mechanism requires an initiator assisted by providing heat treatment.

The ROP process for the synthesis of PDMS consists of initiation, propagation, and termination. The use of an initiator in the form of potassium hydroxide (KOH) which is a strong base will result in an anionic ROP mechanism. A number of KOH with a certain concentration will initiate and form an anion, which represents the active center in the propagation reaction. At the initiation process, the OH ion from KOH donates a pair of electrons to one of the silicon atoms (Si) of the D4 siloxane and binds. As the result, the electron pairs that form siloxane bonds in the monomer ring break from the cyclic chain into a linear chain. The oxygen that gets a pair of electrons with a negative charge will bind to the molecule in the second cyclic monomer, and so on. Termination of the chain occurs due to the use of the end-capping agents, which in this study used disiloxane in the form of hexamethyldisiloxane (MM).

The low viscosity of PDMS using a low grade of D4 monomer coded as Sample A, while low viscosity of PDMS using a high grade of D4 monomer coded as Sample B. Medium viscosity of PDMS coded as Sample C. The synthesis begins with setting the synthesis temperature, then mixing 7.8 mL D4 and 3 mL MM (for volume ratio of D4:MM = 26:10) or 8.9 mL D4 and 1.9 mL MM (for volume ratio of D4:MM = 46:10). After that, a KOH solution with a certain concentration is added. The mixture of these materials is stirred for the specified time to form a gel. Synthesis conditions of all samples are listed in Table 1.

The purification process was carried out by diluting the sample with chloroform. After that, milli-Q water was added. Then the samples are stored until the mixture separates into a liquid and gel phase. Both of them are separated. The pH of the liquid solution was checked to produce a neutral pH value. This purification process was repeated three times. Furthermore, stirring by heating is carried out to remove the chloroform.

PDMS samples were characterized to measure viscosity, refractive index, surface tension, and detect a functional group of the sample. The HET-CAM toxicity test was carried out using seven-day-old fertile white leghorn eggs weighing between 50–60 g. Eggs were tested for test materials (samples A, B, and, C), positive control (1% sodium dodecyl sulfate (SDS)), and negative control (0.9% NaCl). Each substance was tested on three eggs. After that, the eggs will be incubated at a temperature of 38.3 ± 0.2 °C for 10 days. The eggs that have been selected would be disinfected. The shells were removed to expose the membrane of the egg. The membranes were tested against the test material, positive control, and negative control. Observations were made for 300 s by recording the appearance time of each observed endpoint at time intervals of 0 s, 10 s, 30 s, 60 s, 180 s, and 300 s. The evaluation was carried out based on the percentage of the occurrence of the endpoint in the form of hemorrhage, lysis, and coagulation and given a value or scoring as shown in Table 2. The score value is adjusted to the standard reported by the Interagency Coordinating Committee on the Validation of Alternative Methods (ICCVAM) [14].

## 3. Results

### 3.1. PDMS Properties

Table 3 shows the characteristics of viscosity (η), yield, refractive index (n), additional diopters, and surface tension (γ) of all samples. The viscosity value of sample A has a slight difference from sample B. However, both of them are still in the range of low viscosity type. Sample A and sample B are categorized as low viscosity. Meanwhile, sample C has a medium viscosity. All samples have a transparent appearance, as shown in Figure 1.

### 3.2. IR Spectra of PDMS

Functional groups of all samples have been identified and listed in Table 4. Compared with commercial PDMS, all of the samples have slight differences in wavenumber. Nevertheless, the transmittance peaks of all samples show the same spectra and indicate that all of them have the same functional group as the commercial. The infrared (IR) spectra of the samples and commercial are shown in Figure 2.

### 3.3. In Vitro Toxicity Test

The result scores of all samples and reference substances are shown in Figure 3. A significant change happens in the positive control group from 10 s until 300 s. The samples, comparison substances, and negative control did not show a significant change. More complete scoring results for hemorrhage, lysis and coagulation can be seen in Appendix A. Micro images of a blood vessel in the HET-CAM test are shown in Figure 4. Blood vessel damage was seen significantly in the positive control group, while the samples, reference substances, and negative control, did not damage the blood vessel. Detail figures of blood vessel micro images can be seen in Appendix A.

## 4. Discussion

PDMS has been successfully synthesized using a low-grade D4 monomer with different synthesis conditions from PDMS synthesized using a high-grade D4 monomer. Sample A synthesized with higher synthesis temperature and KOH concentration compared to sample B. In addition, the time of polymerization of sample A is also longer than sample B. However, simply by changing the volume ratio of D4 and MM used from 26:10 to 46:10, medium viscosity of PDMS can be produced from the same purity monomer.

Using a low grade of D4 monomer gives some excellent properties, especially for surface tension and yield value of the samples. The high surface tension of sample A and sample C will reduce the possibility of emulsification. However, these samples are still easy to use in vitreoretinal surgery due to their viscosity. The yield value of the samples will affect the number of products produced. The higher the yield values of the samples, the better the production effectiveness. Samples from a low grade of D4 monomer have a higher refractive index than the sample from a high grade of D4 monomer. However, the additional diopters are still in the range of allowable values (+3.0D until +3.5D) [15]. Moreover, sample A and sample C have the appearance as transparent as sample B, as shown in Figure 1.

The IR spectra of all samples showed that all the samples had similar transmittance peaks to commercial but with different intensities. The sample with higher viscosity has lower peak intensity. However, the sample with similar viscosity using a low-grade D4 monomer had higher intensity than sample using a high-grade D4 monomer. The main functional groups of PDMS were found in all samples without any impurities. The FTIR result confirmed that all the samples were PDMS.

The low grade of D4 monomer has successfully produced the good quality of low viscosity and medium viscosity of PDMS. However, if the safety of PDMS is not guaranteed, it will be nothing. For this reason, the toxicological test must be carried out before it is used in the human eyes. Chorioallantoic Membrane (CAM) is a complete tissue in the hen’s eggs and consists of arteries, veins, and capillaries. CAM has a similar reaction process as in rabbit eyes (Draize test), especially in conjunctiva tissue, when it is exposed to an irritating substance. Hen embryos have been used in various fields of medical research since the 20th century. The egg that was used in this test is the fertile white egg. The egg was chosen based on the vascular condition during candling CAM process [11]. Based on interlaboratory validity, which was conducted by Hagino in 1999, the HET-CAM method can provide an alternative method to evaluate potential irritation of chemicals to the eye. CAM test evaluates vascular reactions and damage of the CAM in the presence of macroscopic changes such as hemorrhage, lysis, and coagulation. Previous studies showed that HET-CAM and in vivo eye irritation tests have a good correlation [16].

Compared with other alternative tests, the HET-CAM test has some good points, such as could be carried out for all types of chemical substance (liquid or powder), applied with the similar condition of in vivo test, easy to access for research, not expensive, not need complicated animal room facility, and simple method [10,16]. However, the HET-CAM test is not suitable for red-pigmented samples. To maintain objectivity in the evaluation process of potential eye irritation, the HET-CAM test requires the use of a reference substance. In addition, the HET-CAM test also needs an experienced investigator [16].

Evaluation of potential irritants of the test substance is spelled out with the scoring system. A scoring system has gradually been developed by selecting various criteria to obtain the most consistent correlation with the variation in the concentration of the preparation being tested.

In vitro toxicity test showed that all types of damage occur in positive control. Hemorrhage occurred in all groups starting at 10 s. Lysis occurred from the 30 s. Meanwhile, coagulation occurred from the 60 s. A hemorrhage is a condition when blood comes out of damaged blood vessels. Lysis is when the integrity of the cell membrane is broken or damaged and causes cell organelle to come out. Coagulation is a condition when the blood freezes. Samples were non-irritant through the HET-CAM test. Based on HET-CAM test evaluation, negative control (NaCl 0.9%) was non-irritant, positive control (1% SDS) was a strong irritant, and all PDMS samples get zero (0) score that indicated as non-irritant substances.

Several studies have shown that the low viscosity of PDMS has a higher emulsification tendency than other viscosities. This is influenced by low surface tension and impurities [3]. Both cause ocular toxicity by spreading to the eye tissue. In this study, we successfully repeated the synthesis of low and medium viscosity PDMS with high surface tension. In vitro toxicity tests showed that low-viscosity PDMS from low-grade D4 monomers did not cause irritation, as did PDMS synthesized from high-grade D4 monomers. The results of the PDMS toxicity test with medium viscosity also showed that this PDMS is a non-irritating substance. These results prove that samples with high surface tension values have low emulsification tendencies and reduce sample toxicity. In addition, the use of low-grade D4 monomers can produce high-quality PDMS as well as PDMS synthesized using high-grade D4 monomers.

The result of in vitro toxicity test by the HET-CAM test method is appropriate with the other in vitro toxicity test results. Romano et al. explained that the cytotoxic effect of low viscosity of PDMS, with different purification levels, was not found in human retinal cells (ARPE-19 and BALB 3T3) [5]. Moreover, these in vitro toxicity test results are also suitable within in vivo toxicity tests in rabbit eyes. Mackiewicz et al. explained that low viscosity (1000 mPa.s) and medium viscosity (3000 mPa.s) of PDMS show no signs of inflammation or hyperemia after 3 months tested into rabbit eyes [17].

## 5. Conclusions

We have successfully synthesized low- and medium-viscosity PDMS materials using a low-grade D4 monomer and characterized the physical properties, including the in vitro toxicity test by the HET-CAM test method. The characterization of physical properties showed that the refractive index, surface tension, and functional groups of all PDMS samples were very similar to commercial PDMS and PDMS synthesized using a high-grade D4 monomer. The addition of diopters of PDMS samples is also within the allowed normal range. The results of the in vitro toxicity test using the HET-CAM method showed that PDMS with low and medium viscosity was a non-irritant substance indicating that all samples were non-toxic. These results proved that the use of low-grade D4 monomer can produce high-quality PDMS, is safe, and has the potential to be developed as a substitute for vitreous humor in the human eye.

## Figures and Tables

**Figure 1 jfb-13-00003-f001:**
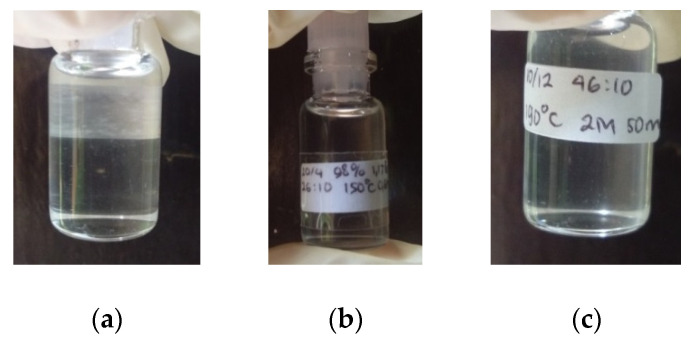
PDMS of (**a**) sample A; (**b**) sample B; and (**c**) sample C.

**Figure 2 jfb-13-00003-f002:**
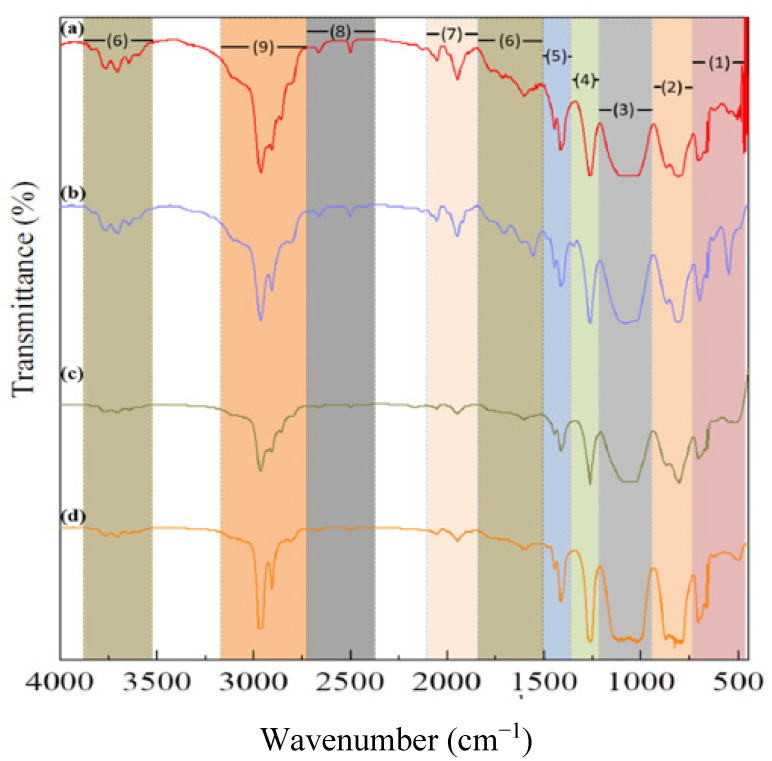
IR spectra of (**a**) sample A; (**b**) sample B; (**c**) sample C; and (**d**) commercial.

**Figure 3 jfb-13-00003-f003:**
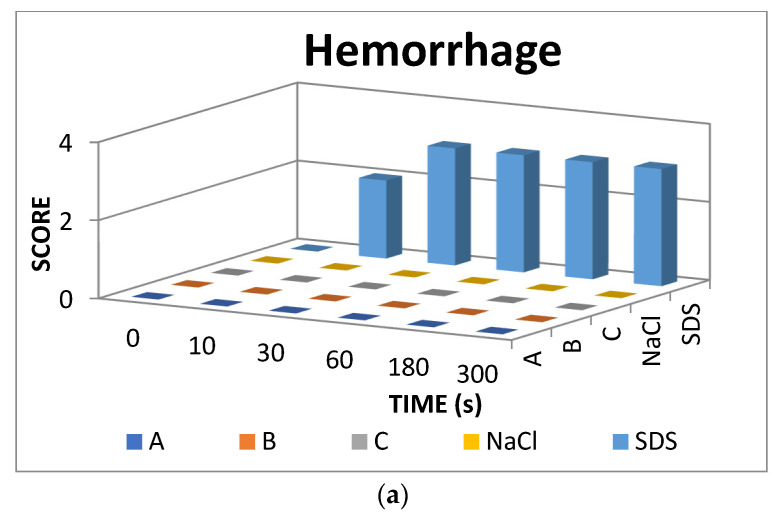
Endpoint score of (**a**) hemorrhage, (**b**) lysis, and (**c**) coagulation.

**Figure 4 jfb-13-00003-f004:**
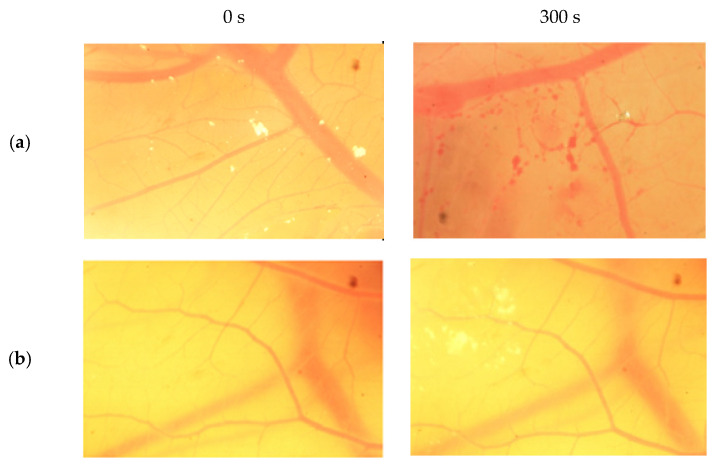
Vessels of sodium dodecyl sulfate (SDS) (**a**), sample A (**b**), sample B (**c**), sample C (**d**), and NaCl (**e**) from 0 s and 300 s.

**Table 1 jfb-13-00003-t001:** Synthesis parameters of the samples.

Condition	A	B	C
Purity of D4 (%)	96	98	96
Ratio of D4:MM	26:10	26:10	46:10
Synthesis Temperature (°C)	190	150	190
KOH Concentration (M)	2	0.6	2
Time of Polymerization (minute)	50	16	50

**Table 2 jfb-13-00003-t002:** Scoring scheme to irritant tests with the Hen’s Egg Test Chorioallantoic Membrane (HET-CAM) method.

Area of the Endpoint (%)	Score
0–10	0
10–30	1
30–60	2
60–100	3

**Table 3 jfb-13-00003-t003:** Polydimethylsiloxane (PDMS) characteristics of η, yield, n, additional diopters, and γ.

Sample	η (Pa.s)	Yield (%)	n	Additional Diopters	γ (mN/m)
A	1.15	67.37	1.4040	3.410	21
B	1.17	54.59	1.3993	3.179	19
C	1.81	71.31	1.4048	3.449	21.5

**Table 4 jfb-13-00003-t004:** Functional group of all samples and commercial PDMS.

No	Functional Group	Wavenumber (cm^−1^)			
		Commercial [13]	A	B	C
1	Si-O-Si	500, 703	523, 700	551, 695	519, 701
2	Si-C stretching and CH_3_ rocking	792, 871	803, 864	807, 863	799, 864
3	Si-O-Si stretching	1112, 1023	1023 1099	1023, 1075	1019, 1093
4	CH_3_ symmetric deformation of Si-CH_3_	1263	1261	1261	1260
5	CH_3_ asymmetric deformation of Si-CH_3_	1412	1413	1412	1412
6	OH	1600, 3643– 3828	1600, 3595– 3823	1600, 3643– 3828	1602, 3593– 3828
7	Si-C	1945, 2052	1945, 2052	1945, 2052	1945, 2052
8	CH	2500, 2663	2500, 2663	2500, 2663	2500, 2663
9	CH stretching of CH_3_	2906, 2972	2906, 2963	2906, 2964	2906, 2964

## Data Availability

Not applicable.

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
