# Peer review of "Physical Characterization and In Vitro Toxicity Test of PDMS Synthesized from Low-Grade D4 Monomer as a Vitreous Substitute in the Human Eyes"

_jfb, 2022, doi:10.3390/jfb13010003_

Round 1

Reviewer 1 Report

The manuscript of an article presented by Auliya, Risdiana and co-workers, describes the synthesis of PDMS 2 Synthesized from Low Grade D4 Monomer and their properties towards Substitution for vitreous humour in human eyes.

Some parts need to be verified and corrected.

What was the aim of the conducted studies? It is not clear to me. Line 71-73 There seem to be the hypothesis of the research to be conducted. In my opinion, it should be highlighted more and respectively presented in conclusion.

The synthetic procedure for PDMS is ROP is described in a very narrow way.

Is there any pattern for the conditions changes? Could you describe more in this aspect?

What are the results in regards to the obtained DPn, PDI, Mn, Mw. There is no information on this for obtained samples A, B, C. Is there any NMR analysis performed? All these results should be placed in Supplementary Material.

Figure 1 – what is the difference in respective samples A, B, C? The way they are presented right now exhibits no significant difference. If so, what is the point to show them?

Figure 2. The figure caption should be placed below it, not above.

Table 4. The bands should be rather presented in integers. Please add information on the wavenumbers [cm-1].

Figure 3, 4, 5. The respective figures a, b… etc. are not visible from their right side, they are seemed to be cut as point D is not visible. Please verify.

Comment to Figure 3, 4, 5 a and also Figure 6-10– could you please place only the most referenced chart in the manuscript and the rest in Supplementary Material.   

Please verify the editorial and language part of the manuscript before submitting a revision.

Author Response

Response to Reviewer 1 Comments

We are grateful for suggestions and comments submitted to improve the quality of our manuscripts. Here is the response to the reviewer's comments:

Point 1: What was the aim of the conducted studies? It is not clear to me. Line 71-73 There seem to be the hypothesis of the research to be conducted. In my opinion, it should be highlighted more and respectively presented in conclusion.

Response 1: Our study focuses on how to produce high quality PDMS from low-grade D4 monomer and reported for the first time the in vitro toxicity test of these PDMS samples using the HET-CAM test method to obtain information safety and toxicity level of all PDMS samples. The clear statement of the aim of this study is added in introduction and conclusion.

Point 2: The synthetic procedure for PDMS is ROP is described in a very narrow way.

Response 2: We already revised the materials and method section. We added more information about ROP procedure including detail information about synthesis method.

Point 3: Is there any pattern for the conditions changes? Could you describe more in this aspect?

Response 3: When conditions change, for example when carrying out synthesis using low grade D4 monomer, PDMS was successfully synthesized with higher synthesis temperature and KOH concentration, as well as a longer synthesis time compared to using high grade D4 monomer. In addition, simply by changing the volume ratio of D4 and MM used from 26:10 to 46:10, higher viscosity PDMS can be produced from low grade D4 monomer. However, using either low grade or high grade D4 monomer both results in high quality PDMS.

Point 4: What are the results in regards to the obtained DPn, PDI, Mn, Mw. There is no information on this for obtained samples A, B, C. Is there any NMR analysis performed? All these results should be placed in Supplementary Material.

Response 4: In this study we focused on the synthesis and characterization of the toxicity of the material. According to the European Chemical Agency (ECHA), D4 monomer is a toxic substance. Therefore, products produced from D4 monomer may contain hazardous and toxic substances. We succeeded in proving that our synthesis process was running well marked by the absence of signs of the polymer being toxic, so in this study we did not measure NMR because the polymer we produced was proven to be non-toxic. We plan to carry out NMR measurements for future studies. We have revised the introduction by adding an explanation of the importance of the toxicity test because the monomers used are toxic.

Point 5: Figure 1 – what is the difference in respective samples A, B, C? The way they are presented right now exhibits no significant difference. If so, what is the point to show them?

Response 5: We show Figure 1 which is a photo of samples A, B and C is to show a transparent physical appearance on all samples, both synthesized using low grade and high grade D4 monomer. Samples A, B and C showed the same appearance indicating that samples synthesized using low grade D4 monomer produced samples with good transparency.

Point 6: Figure 2. The figure caption should be placed below it, not above.

Response 6: We have revised Figure 2 by removing the figure caption at the top and correcting the numbering range of the image.

Point 7: Table 4. The bands should be rather presented in integers. Please add information on the wavenumbers [cm-1].

Response 7: We have revised the FTIR data and modified the bands to be integers. We have also added unit information in the wavenumber section of the Table 4.

Point 8: Figure 3, 4, 5. The respective figures a, b… etc. are not visible from their right side, they are seemed to be cut as point D is not visible. Please verify.

Response 8: We have revised Figure 3, 4, and 5. Figure 3, 4 and 5 are combined into one figure, namely Figure 3. We revised the figure so that the vertical axis on the right side becomes visible.

Point 9: Comment to Figure 3, 4, 5 a and also Figure 6-10– could you please place only the most referenced chart in the manuscript and the rest in Supplementary Material.

Response 9: We have revised Figure 3-10. For Figure 3, 4, and 5 we collect the most important information from the three figures and display it in one figure. Similarly for Figure 6-10, we collect important information and display it on a single figure. Other figures as a result of more detailed HET-CAM testing are shown in the Supplementary Material section.

Reviewer 2 Report

This manuscript re-produced low/medium-viscous PDMSs by previously reported method and studied its biocompatibility by HET-CAM test. I didn't see any novalty. Especially, authors mentioned that the low molecular weight by-products can cause ocular toxicity, which are considered as D4, D5 and D6. However, they didn't well characterize whether the synthesized PDMSs contained those low molecular monomers. I didn't see any proofs, such as 1H and 29Si NMR, LC-MS, GPC, etc. Authors should provide the strong evidences that no low molecular cyclic monomers existed after proper purifications. Also ther e is no detailed synthesis procedures among different monomers.

Furthermore, there are lots of significant differences on the FTIR spectra between sample A/B and commercial one, including ~3750, 1500-1750, 500-700 cm-1, not slight differences. Authors should give detailed explanations and further characterizations.

Overall, I didn't recommend publication.

Author Response

Response to Reviewer 2 Comments

We are grateful for suggestions and comments submitted to improve the quality of our manuscripts. Here is the response to the reviewer's comments:

Point 1: This manuscript re-produced low/medium-viscous PDMSs by previously reported method and studied its biocompatibility by HET-CAM test. I didn't see any novalty. Especially, authors mentioned that the low molecular weight by-products can cause ocular toxicity, which are considered as D4, D5 and D6. However, they didn't well characterize whether the synthesized PDMSs contained those low molecular monomers. I didn't see any proofs, such as 1H and 29Si NMR, LC-MS, GPC, etc. Authors should provide the strong evidences that no low molecular cyclic monomers existed after proper purifications. Also ther e is no detailed synthesis procedures among different monomers.

Response 1: For novelty we add a few sentences that clearly mention the novelty of our research. The novelty of this study is that we were able to obtain information about PDMS synthesized from low grade D4 monomer show non-toxic tested by HET-CAM method. In vitro toxicity test is very important because according to the European Chemical Agency (ECHA), D4 monomer is a toxic substance. Therefore, products produced from D4 monomer may contain hazardous and toxic substances. We succeeded in proving that our synthesis process was running well marked by the absence of signs of the polymer being toxic, so in this study we did not measure NMR because the polymer we produced was proven to be non-toxic. We have revised the introduction by adding an explanation of the importance of the toxicity test because the monomers used are toxic.

We also already revised the materials and method section. We added more information about ROP procedure including detail information about synthesis method.

Point 2: Furthermore, there are lots of significant differences on the FTIR spectra between sample A/B and commercial one, including ~3750, 1500-1750, 500-700 cm-1, not slight differences. Authors should give detailed explanations and further characterizations.

Response 2: We have revised the Figure 2 of FTIR spectra. We have added identification for all visible peaks such as peaks of ~3750, 1500-1750, 500-700 cm-1. We also revised the numbering of the FTIR peaks according to the wavenumber range. Apart from adding the identity of the peaks observed from the FTIR measurement results, we also added an analysis for the change in intensity for each observed peak. We added an explanation of the reasons for the difference in intensity at certain peaks in the Discussion section.

Round 2

Reviewer 1 Report

What is the toxicity of the D4 monomer? Could You provide any relevant information on the measure of its toxicity?

Is there any NMR analysis of obtained PDMS performed that could be uploaded in the Supplementary Information file?

Author Response

Response to Reviewer 1 Comments

We are grateful for suggestions and comments submitted to improve the quality of our manuscripts. Here is the response to the reviewer's comments:

Point 1: What is the toxicity of the D4 monomer? Could You provide any relevant information on the measure of its toxicity?

Response 1: In the eye, the D4 monomer is toxic by diffusing into the ocular tissue and causing severe inflammation. The D4 monomer has been reported to cause severe corneal edema and opacification. We added some information about the results of the D4 monomer ocular toxicity test in the Introduction section. We also added references that explain that the D4 monomer has toxic properties, namely reference numbers 6 and 7.

Point 2: Is there any NMR analysis of obtained PDMS performed that could be uploaded in the Supplementary Information file?

Response 2: In this research, we did not measure NMR because of the limitations of suitable instruments in Indonesia. However, the information about the functional groups and toxicity test of our samples which we describe in this manuscript is appropriate to describe the quality of the PDMS product. We hope that information on the NMR structure of the PDMS can be reported in our next publication.

Reviewer 2 Report

The authors have answered all my questions and this manuscript in its present form has presented sufficient data and evidences to show the physical properties and in vitro toxicity of PDMS synthesized from low grade D4 monomers. Although the toxic tests have proved that the synthesized PDMS is biocompatible, I still consist that the authors should clearly know the chemical compositions of the synthesized PDMSs from low degree D4 compared with the commercialized ones. Detailed NMR and GPC data should give readers more confidences. Anyway, the key evidence of toxicity proved effectively. I consider that it could be accepted as a research article in the present state.

Author Response

Response to Reviewer 2 Comments

Point 1: The authors have answered all my questions and this manuscript in its present form has presented sufficient data and evidences to show the physical properties and in vitro toxicity of PDMS synthesized from low grade D4 monomers. Although the toxic tests have proved that the synthesized PDMS is biocompatible, I still consist that the authors should clearly know the chemical compositions of the synthesized PDMSs from low degree D4 compared with the commercialized ones. Detailed NMR and GPC data should give readers more confidences. Anyway, the key evidence of toxicity proved effectively. I consider that it could be accepted as a research article in the present state.

Response 1: We are grateful for suggestions and comments submitted to improve the quality of our manuscripts. In this research, we did not measure NMR and GPC because of the limitations of suitable instruments in Indonesia. However, the information about the functional groups and toxicity test of our samples which we describe in this manuscript is appropriate to describe the quality of the PDMS product. We hope that information on the chemical composition of the PDMS can be reported in our next publication.
